# Changes in the Density and Composition of Rhizosphere Pathogenic *Fusarium* and Beneficial *Trichoderma* Contributing to Reduced Root Rot of Intercropped Soybean

**DOI:** 10.3390/pathogens11040478

**Published:** 2022-04-16

**Authors:** Huiting Xu, Li Yan, Mingdi Zhang, Xiaoli Chang, Dan Zhu, Dengqin Wei, Muhammd Naeem, Chun Song, Xiaoling Wu, Taiguo Liu, Wanquan Chen, Wenyu Yang

**Affiliations:** 1Department of Plant Protection, College of Agronomy & Sichuan Engineering Research Center for Crop Strip Intercropping System, Sichuan Agricultural University, Chengdu 611130, China; m1349718692@163.com (H.X.); AgricultureZD@163.com (D.Z.); dengdenger10@163.com (D.W.); muhammdnaeem201@gmail.com (M.N.); songchun@sicau.edu.cn (C.S.); wulx@sicau.edu.cn (X.W.); mssiyangwy@sicau.edu.cn (W.Y.); 2State Key Laboratory for Biology of Plant Diseases and Insect Pests, Institute of Plant Protection, Chinese Academy of Agricultural Sciences, Beijing 100193, China; mirror_dis@126.com (L.Y.); liutaiguo@caas.cn (T.L.); wqchen@ippcaas.cn (W.C.); 3Department of International Law Affairs, Dong-a University, Busan 49236, Korea; yxxy66@163.com

**Keywords:** maize–soybean relay strip intercropping, *Fusarium* root rot, *Trichoderm* spp., *Fusarium* spp., antagonism

## Abstract

The dynamic of soil-borne disease is closely related to the rhizosphere microbial communities. Maize–soybean relay strip intercropping has been shown to significantly control the type of soybean root rot that tends to occur in monoculture. However, it is still unknown whether the rhizosphere microbial community participates in the regulation of intercropped soybean root rot. In this study, rhizosphere *Fusarium* and *Trichoderma* communities were compared in either healthy or root-rotted rhizosphere soil from monocultured and intercropped soybean, and our results showed the abundance of rhizosphere *Fusarium* in intercropping was remarkably different from monoculture. Of four species identified, *F. oxysporum* was the most aggressive and more frequently isolated in diseased soil of monoculture. In contrast, *Trichoderma* was largely accumulated in healthy rhizosphere soil of intercropping rather than monoculture. *T. harzianum* dramatically increased in the rhizosphere of intercropping, while *T. virens* and *T. afroharzianum* also exhibited distinct isolation frequency. For the antagonism test in vitro, *Trichoderma* strains had antagonistic effects on *F. oxysporum* with the percentage of mycelial inhibition ranging from 50.59–92.94%, and they displayed good mycoparasitic abilities against *F. oxysporum* through coiling around and entering into the hyphae, expanding along the cell–cell lumen and even dissolving cell walls of the target fungus. These results indicate maize–soybean relay strip intercropping significantly increases the density and composition proportion of beneficial *Trichoderma* to antagonize the pathogenic *Fusarium* species in rhizosphere, thus potentially contributing to the suppression of soybean root rot under the intercropping.

## 1. Introduction

Soybean root rot is one of the detrimental soil-borne fungal diseases in soybean production worldwide. A variety of pathogens, including *Fusarium*, cause soybean root rot, leading to 10–60% loss of soybean yield [1,2,3]. This disease is remarkably affected by cropping pattern [4,5,6,7]. In Northeast China, *Fusarium* root rot is very popular because of long-term continuous soybean monoculture and has been recognized as one important limited factor of soybean production [6]. With the increasing of soybean planting area in Southwest China, this disease has potentially threatened local soybean production [7,8]. Currently, the control of soybean root rot is mainly dependent on agricultural practices such as crop rotation or intercropping, soil tilling, varieties with effective and durable resistance as well as seed treatment [5,9,10]. Although resistance breeding of soybean has largely developed with modern molecular techniques, cost-effective and durable resistant varieties are still lacking for application [4,5,11,12]. Chemicals are very helpful to prevent pathogen infection, but often fail to cure a plant once infection occurs, and otherwise sometimes affect beneficial soil microbes due to excessive accumulation in the soil environment [13,14]. Biological control has always attracted considerable attention to the sustainable management of soybean root rot because of its advantages on the balance of crop production and agricultural environment protection [15,16,17].

*Trichoderma* is one genus of fungi in the family Hypocreaceae, which is commonly existing in soil, root and foliar ecosystem [18]. *T. lignorum* was firstly reported in 1983 to have an antagonistic effect on *Rhizoctonia solani* [19]. Many *Trichoderma* species, such as *T. harzianum*, *T. virens*, *T. asperellum*, *T. hamatum*, *T. longibrachiatum* and *T. koningii*, have successively been identified as antagonistic fungi against plant pathogens so far [20,21], especially soil-borne pathogenic fungi [22,23,24,25]. For example, *T. harzianum* and *T. asperellum* are commercially effective biological control agents against *F. oxysporum* causing watermelon wilt [26], and *T. asperellum* isolates significantly suppress *Fusarium* wilt of tomato [27]. Some *Trichoderma* strains can confer biocontrol either directly by interacting with pathogens via mycoparasitism, or by competition for nutrients or root niches, while other strains establish robust and durable colonization of root surfaces and penetrate into the epidermal cells to indirectly induce the host resistance and enhance root growth [20,28]. Accordingly, *Trichoderma* has become one of the most extensively studied beneficial fungi for agricultural crop improvement [29], and a few species have already been explored as fungal biocontrol agents (BCA) or bio-pesticides for the biocontrol of the soil-borne fungal diseases [30]. 

Research increasingly indicates that the cropping pattern can affect the soil microbial community, change the composition of beneficial and pathogenic microbes and regulate the occurrence of soil-borne disease [31]. In the *Radix pseudostellariae* rhizosphere, the consecutive monoculture increases the abundance of the pathogenic *F. oxysporum* but decreases *Trichoderma* spp. [32]. Recently, a maize–soybean relay strip intercropping has been widely practiced in Southwest China, and this cropping pattern is characterized by two-row maize plant strips intercropped with two to four rows of soybean which have positive effects on increasing the land equivalent ratio [33], improving the soil nutrients and microbe structure [34,35,36], reducing weeds and diseases in the field [7,37] and increasing crop yield [38,39] as compared to soybean monoculture. This intercropping can also reduce the disease severity of *Fusarium* root rot and change the population diversity of the pathogenic *Fusarium* species in diseased soybean roots [7]. However, the underlying mechanism of regulating soybean root rot by this intercropping is largely unclear. 

In the current study, we will focus on rhizosphere microbial communities and their participation in the regulation of soybean root rot in maize–soybean relay stripping intercropping as compared to soybean monoculture. For this, the aims were: 1—the density of *Fusarium* and *Trichoderma* communities was identified and compared from diseased soybean rhizosphere soil and healthy soybean rhizosphere soil of intercropping and monoculture; 2—the antagonistic effects of *Trichoderma* species on the pathogenic *F. oxysporum* of soybean root rot were examined in vitro. This study will be meaningful for uncovering the rhizosphere microbial regulation of maize–soybean relay strip intercropping on *Fusarium* root rot and exploring the beneficial *Trichoderma* strains for the biocontrol of soil-borne diseases.

## 2. Results

### 2.1. Population Density of Rhizosphere Fusarium and Trichoderma in Response to Intercropping and Monoculture

To uncover the influence of cropping patterns on the density of rhizosphere *Fusarium* and *Trichoderma* communities associated with soybean root rot, the richness of these two fungi was compared between diseased rhizosphere soil and healthy rhizosphere soil, as well as between intercropping and monoculture. As shown in Table 1, *Fusarium* was the richest with 1046.51 cfu·g^−1^ in diseased rhizosphere soil of intercropping (IDR), whereas it had no significant difference in two soil types of soybean monoculture. In contrast, *Trichoderma* richness in two soil types of intercropping (IDR and IHR) was much higher than that of monoculture (MDR and MHR). Meanwhile, *Trichderma* was also much richer in healthy rhizosphere soil (IHR) than diseased rhizosphere soil (IDR) of maize–soybean relay strip intercropping, but it had almost no difference between two soil samples of soybean monoculture. Thus, it is clear that *Fusarium* richness was significantly declined in healthy rhizosphere soil in two cropping patterns followed by the increasing of *Trichoderma*, and rhizosphere *Trichoderma* richness is more dramatically affected by cropping pattern while *Fusarium* richness is also remarkably distinct with respect to the occurrence of root rot under the intercropping.

### 2.2. Species Identification of Rhizosphere Trichoderma and Fusarium in Response to Intercropping and Monoculture

In this study, a total of 165 *Fusarium* isolates were obtained from four soil types of intercropping and monoculture. Sequences of *EF-1α* and *RPB2* genes were amplified using primer pairs (Appendix A) and analyzed by blasting on the *Fusarium* MLST database, which showed that these isolates shared more than 98% sequence identities with those reference isolates of *F.solani* species complex (FSSC), *F.oxysporum*, *F.incarnatum-equiseti* species complex (FIESC) and *F.commune* (Appendix A), respectively. A phylogenetic tree based on *EF-1α* and *RPB2* was constructed with the representative isolates, reference isolates from NCBI, and the outgroup isolate *Nectriaceae* sp. (NRRL52754) (Appendix A), and the representative isolates from intercropping were absolutely classified into three clades including FSSC, FIESC and *F. oxysporum* in this tree (Figure 1A), while those isolates from monoculture were claded into FSSC, *F. oxysporum* and *F. commune*, respectively (Figure 1B), thus displaying a distinct species composition of *Fusarium* between intercropping and monoculture.

*Trichoderma* isolates were recovered from rhizosphere soil of diseased and healthy soybean plants in two cropping patterns. Blastn analysis of partial *ITS*, *EF-1α* or *RPB2* genes on the NCBI database showed that these isolates had more than 95% sequence identities with *T. harzianum, T. virens* and *T. afroharzianum* (Appendix A). For phylogenetic analysis (Figure 2), all representative isolates from either intercropping or monoculture were clearly claded intro *T. harzianum*, *T. virens* and *T. afroharzianum*, respectively, in the *ITS*/*RPB2*-based MP trees, whereas *T. harzianum* and *T. afroharzianum* were not totally discriminated in the *ITS/EF-1α*-based phylogenetic trees in both cropping patterns (Appendix A). Thus, *Trichoderma* isolates from rhizosphere soil were identified as *T. virens*, *T. afroharzianum* and *T. harzianum*, and there was no difference in species diversity between intercropping and monoculture.

### 2.3. Composition and Isolation Proportion of Fusarium spp. and Trichoderma spp. Affected by Intercropping and Monoculture 

The composition and isolation proportion of rhizosphere *Fusarium* and *Trichoderma* spp. was further compared between monoculture and intercropping as well as between diseased and healthy soils. As shown in Figure 3A,B, FSSC had the highest isolation proportion in all soil types that reached up to 88.89% (40/45) of *Fusarium* composition and 24.24% (40/165) of all isolated *Fusarium* in IDR as compared to 65% (28/43) and 16.97% (28/165) in MDR (Figure 3A,B), respectively. *F. oxysporum* accounted for the second largest proportion in different soil samples and had the highest proportion of 32.56% (14/43) in MDR than that of 8.89% (4/45) in IDR (Figure 1A). The FIESC and *F. commune* were specifically isolated species in intercropping and monoculture, respectively, with relatively less proportion in corresponding soil samples (Figure 1A,B).

In this study, a total of three *Trichoderma* species were obtained from rhizosphere soils of two cropping patterns, but the species composition and isolation proportion varied over the occurrence of root rot as well as cropping pattern (Figure 3C,D). Both *T. harzianum* and *T. virens* were isolated from all soil types, but *T. harzianum* was the most predominant species of *Trichoderma* (Figure 3D). The composition proportion of *T. harzianum* was 95.9 % (71/74) and 76.2% (32/42) in IHR and IDR, whereas it was as low as 66.7% (6/9) and 33.3% (3/9) in MHR and MDR, respectively. As the secondary dominant species, *T. virens* of *Trichoderma* composition was clearly declined in two soil types of intercropping when compared with monoculture, whereas an increase in *T. afroharzianum* appeared in MHR and MDR of monoculture and in IDR of intercropping. These results indicate that intercropping changes *Trichoderma* composition and remarkably increases *T. harzianum* isolation proportion in comparison to monoculture. 

### 2.4. The Pathogenicity of Rhizosphere Fusarium spp. on Soybean

Pathogenicity test of rhizosphere *Fusarium* species on soybean seedlings showed that all four *Fusarium* species were able to infect soybean roots and cause stunted, brown, rotted taproots and hair roots, but displayed significantly different aggressiveness (Figure 4A). *F. oxysporum* was the most aggressive species among all *Fusarium* species with a disease severity index (DSI) ranging from 35 to 90 (Figure 4B). The DSI of *F. oxysporum* isolates from monoculture were much higher than those from intercropping, and the highest DSI went up to 90 (Figure 4B) followed by serious growth inhibition and rotted taproot (Figure 4A). FSSC as the most isolated fungi had moderate aggressiveness with DSI around 50, and there was no significant difference among those isolates from two cropping patterns. In addition, *FIESC* and *F. commune* separately for intercropping and monoculture were also moderately pathogenic to soybean, but different for their aggressiveness.

### 2.5. Inhibition Effects of Trichoderma spp. on the Pathogenic F. oxysporum of Soybean Root Rot

As compared to the *F. oxysporum* control, the representative isolates of three *Trichoderma* species grew towards *F. oxysporum* and significantly inhibited mycelial growth of *F. oxysporum* through spatial competition on PDA confrontation culture plates (Figure 5A), and the inhibition response had almost no difference among the same *Trichoderma* species from intercropping and monoculture. Furthermore, the percentage of mycelial inhibition (PMI) was distinct among three *Trichoderma* species, but 38.10% of isolates had the PMI values in the range of 60–70% (Figure 5B). *T. harzianum* isolates showed almost the same range of PMI between intercropping and monoculture, whereas *T. afroharzianum* and *T. virens* had slightly different central PMI values for tested isolates from two cropping patterns. Under two cropping patterns, there was almost no difference in the mean PMI of *T. harzianum*, which were 58.91% in intercropping and 57.35% in monoculture, respectively. The mean PMI for *T. virens* was concentrated at 72.0% in intercropping, but it was as high as 60.30% in monoculture (Figure 5C). In contrast, the mean PMI for *T. afroharzianum* was slightly higher, up to 73.14% in intercropping as compared to 66.18% in monoculture (Figure 5C). This indicates that *T. virens* and *T. afroharzianum* might be also related to *F. oyxsporum* inhibition during rhizosphere fungal interaction with respect to cropping pattern.

To observe the hyphae interaction, *T. harzianum* grew towards (Figure 6A), attached to the hyphae of *F. oxysporum*, and coiled around them (Figure 6B). The strains TRB1-15 and TRB1-7 of *T. virens* coiled around the hyphae, formed brown halos at the contact sites (Figure 6C), and some of them directly entered into the lumen of the target fungus (Figure 6D). As shown in Figure 6E, the hyphae of *T.afroharzianum* entered into the cell lumen of *F. oxysporum* and expanded along the host hyphae (Figure 6E), and resulted in dissolution of cell walls (Figure 6F). In general, rhizosphere *Trichoderma* strains display good mycoparasitic abilities against the pathogenic *F. oxysporum* of soybean root rot.

## 3. Discussion

Rhizosphere soil microbes play an important role in regulating soil-borne diseases that are closely related with cropping patterns [40,41,42]. Our previous study proved that maize–soybean relay strip intercropping suppressed the occurrence of soybean root rot and changed the diversity of the pathogenic *Fusarium* species [7]. In this study, we further focused on this intercropping effect on rhizosphere *Fusarium* communities and potential biocontrol *Trichoderma* communities in association with soybean root rot. We found that continuous maize–soybean relay strip intercropping caused a significant accumulation of rhizosphere *Trichoderma*, but reduced the abundance of the pathogenic *Fusarium* communities, in particular, in healthy soils. In contrast, consecutive soybean monoculture displayed almost no influence on the density of *Trichoderma* and *Fusarium*, no matter whether in diseased or healthy rhizosphere soils. These findings demonstrate that maize–soybean relay strip intercropping is beneficial to induce the accumulation of the rhizosphere *Trichoderma* community. 

*Fusarium* species are the dominant pathogens of soybean root rot in Sichuan Province, Southwest China [7], and these *Fusarium* species often can survive saprophytically and accumulate largely on crop debris, even on or inside cultivated soil, thus serving as primary inocula for soybean infection in rhizosphere microenvironments in the following epidemic year [43]. In our study, four *Fusarium* species including *F. oyxsporum*, FSSC, FIESC and *F. commune* were identified from all rhizosphere soil samples and all resulted in pathogenicity in soybean. Among them, FSSC and *F. oxysporum* were most frequently isolated in both cropping patterns, which is similar to those by Liu et al. [44] who isolated *Fusarium* spp. from soybean rhizosphere soil of different rotation systems in the black soil area of Northeast China. However, *F. oxysporum*, rather than FSSC, had the highest isolation proportion from soybean rhizosphere soil in the black soil area [44], indicating an association between the difference in dominant rhizosphere *Fusarium* species and soybean ecological planting areas. Our previous study reported FSSC and *F. oxysporum* were the predominant species isolated from diseased soybean root in both maize–soybean relay strip intercropping and soybean monoculture [7,8]. In contrast, in this study, the diversity of *Fusarium* spp. was remarkably lower from diseased and healthy rhizosphere soils of two cropping patterns. These results indicate that some *Fusarium* species causing soybean root rot, such as *F. gramniearum*, *F. asiaticum*, *F. proliferatum*, might come from aboveground inoculums instead of rhizosphere soil-borne pathogens. Moreover, FIESC and *F. commune* were specific to intercropping and monoculture with relatively low isolation proportion, respectively, which is basically consistent with our previous findings that specifically isolated *F. commune* and a low isolation proportion of FIESC from the diseased soybean roots in monoculture rather than maize–soybean relay strip intercropping [7]. In addition, compared with intercropping, soybean monoculture did not change the density of rhizosphere *Fusarium*, but dramatically raised the proportion of composition and isolation of the most aggressive *F. oxysporum*. Therefore, unlike intercropping, more severe root rot in soybean monoculture might be related to a relatively higher proportion of the aggressive *F. oxysporum* but not a higher density of *Fusarium* communities. This can be supported by another study that continuous soybean cropping increased the population of *Fusarium* and tended to increase the susceptibility to root rot [45].

Beneficial soil microbes have some advantages on suppressing soil-borne pathogenic fungi, promoting plant growth or decomposing plant residues [46]. *Trichoderma* has widely been recognized as biological alternatives of soil-borne diseases in crop production [24,25], but *Trichoderma* communities are often affected by cropping patterns [47]. Previous studies found that compared with soybean–maize rotation, continuous soybean monoculture decreased the populations of *Trichoderma* [47]. In contrast, the abundance of soil *Trichoderma* increased significantly under the intercropping of sorghum and soybean [48]. Wei et al. (2014) previously identified three *Trichoderma* species, mainly *T. harzianum* and *T. virens*, and a small amount of *T. viride*, from soybean rhizosphere soil for different rotation years in Heilongjiang Province, China. Similarly, we found that *T. harzianum* had the highest isolation proportion from soybean rhizosphere soil of two cropping patterns, followed by *T. virens* and *T. afroharzianum.* We also observed that these *Trichoderma* species displayed a good mycoparasitic abilities in the antagonist against the pathogenic *F. oxysporum* of soybean root rot, and they interacted with *F. oxysporum* mainly through coiling around the hyphae, entering into the lumen of the target fungus, expanding into the cell–cell area and finally dissolving the cell wall of the host fungus. Thus, these antagonistic interactions of *Trichoderma* and *Fusarium* might explain that with the increase in *Trichoderma*, the *Fusarium* community was significantly decreased in healthy rhizosphere soil than that in diseased rhizosphere soil under intercropping pattern. In other studies, *T. virens* has been recognized as an aggressive mycoparasite that is capable of competing with pathogens at the site of infection [49] and also induces JA- and SA-mediated tomato resistance against *Fusarium* wilt [50]. From this point of view, *Trichoderma* communities are an important group in response to maize–soybean relay strip intercropping as compared to soybean monoculture. This is also supported by Chen et al. [32] who expanded that the monoculture of *R. pseudostellariae* altered *Trichoderma* communities in the plant rhizosphere leading to relatively low level of antagonistic microorganisms, and *T. harzianum* ZC51 could inhibit the pathogenic *F. oxysporum* and induce the expression of defense genes. However, several studies have shown that *Trichoderma* strains with a good plate-confrontation effect on pathogens might be not necessarily good, or even no effect, which is predicted to associate with the colonization ability of *Trichoderma* on plants [51]. Therefore, the antagonistic mechanism of these *Trichoderma* strains against *F. oxysporum* causing soybean root rot should be further verified in the actual field production. 

## 4. Materials and Methods

Maize–soybean relay strip intercropping and soybean monoculture were continuously planted since 2012 at Yucheng District, Yaan City, China (29°59″3.17′ N, 102°59″2.57′ E) as described by Chang et al. [7]. This field experiment was designed using the randomized complete block for three replicated experimental plots. Maize cultivar “Denghai605” and soybean cultivar “Nandou12” were selected in the intercropping or monoculture, and all plots were not tilled before sowing to remain soil microbial community. 

### 4.1. Collection of Soil Samples

At the R2 growth stage of soybean in the summer of 2018, soil samples were collected from diseased soybean plants displaying root rot and healthy soybean plants in each experimental plot of intercropping (marked as IDR and IHR) and monoculture (marked as MDR and MHR), respectively. The rhizosphere soil was carefully scraped from soybean root hair using a hairbrush after shaking off the loosely attached soil from the soil block. The soil of 9 plants accounting for about 5% of total plants in each plot were mixed as one sample and kept in an icebox with a sterile polyethylene bag, and then were used for the isolation and identification of *Fusarium* and *Trichoderma* species.

### 4.2. Isolation and Purification of Fusarium and Trichoderma

After drying and filtering using a 60 μm sieve, *Fusarium* were isolated from soil samples using sterilized *Fusarium*-selective medium (PPA, 5 g∙L^−1^ Tryptone, 1.0 g∙L^−1^ KH_2_PO_4_, 0.5 g∙L^−1^ MgSO_4_, 1.0g∙L^−1^ quintozene, 20.0 g∙L^−1^ agar) referred to previous methods [44]. Rhizosphere soil with different weights (0.020 g, 0.025 g, 0.030 g, 0.035 g, 0.040 g, 0.045 g and 0.050 g) were mixed with PPA medium poured on 90 mm plates and then incubated at 25 °C for 5–7 days. The colony number on PPA plates with the proper soil weight was calculated. Pure isolates of *Fusarium* were obtained through single-spore isolation method and transferred into Potato Dextrose Agar (PDA, 200 g∙L^−1^ potato, 10 g∙L^−1^ glucosum anhydricum and 15 g∙L^−1^ agar) for further analysis. 

Isolation of soil *Trichoderma* was conducted using dilution plate method using *Trichoderma*-selective medium (TSM, glucose 3 g∙L^−1^, monobasic potassium phosphate 0.9 g∙L^−1^, magnesium sulfate 0.2 g∙L^−1^, 1/3000 rose-bengal 100 mL∙L^−1^, potassium chloride 0.15 g∙L^−1^, ammonium nitrate1 g∙L^−1^, agar 20 g∙L^−1^) as described by Elad et al. [52] with minor revisions. Total of 10 g of soil samples was suspended in 90 mL sterilized water containing 0.5% Tween−80 and shaken at 25 °C, 150 r·min^−1^ for 40 min. About 1 mL of soil suspension was then diluted into 10^−1^, 10^−2^, 10^−3^, 10^−4^, and 10^−5^ with sterilized water. Additionally, 1 mL diluted soil suspension was mixed with TSM medium and spread on 9 cm Petri plates. All culture plates were incubated at 25 °C in the darkness for 3–5 days. Colony number of *Trichoderma* in different diluted soil suspensions was calculated. For purification, the single spore of the putative *Trichoderma* colonies was transferred on PDA plates and cultured at 25 °C. 

### 4.3. Richness of Rhizosphere Fusarium and Trichoderma

The richness of *Fusarium* and *Trichoderma* in different soil samples was calculated using water content (%) and colony concentration (cfu∙g^−1^) on proper diluted soil suspension by the formula below (1) and (2).
(1)$$\mathrm{soil}\;\mathrm{water}\;\mathrm{content}\;(\%)=\frac{\mathrm{fresh}\;\mathrm{soil}\;\mathrm{weight}-\mathrm{dried}\;\mathrm{soil}\;\mathrm{weight}}{\mathrm{dried}\;\mathrm{soil}\;\mathrm{weight}}\times 100$$
(2)$$\mathrm{colony}\;\mathrm{units}(\mathrm{cfu}\cdot g^{-1}\mathrm{dried}\;\mathrm{soil})=\frac{\mathrm{colony}\;\mathrm{number}\;\mathrm{per}\;\mathrm{plate}\times \mathrm{dilution}\;\mathrm{fold}}{1-\;\mathrm{soil}\;\mathrm{water}\;\mathrm{content}}$$

### 4.4. DNA Extraction and PCR Amplification

The mycelia of fungal isolates were collected from 7-day-old isolates on PDA dishes and then ground in liquid nitrogen with a disposable pellet pestle. Total genomic DNA of all isolates was extracted using a SP Fungal DNA Kit (Aidlab Biotech, Chengdu, China) according to the manufacturer’s protocols. The partial sequences of translation elongation factor 1-alpha (*EF-1α*) and RNA polymerase beta large subunit II (*RPB2*) were amplified for the identification of *Fusarium* species using the primer pairs EF1/EF2 [53] and RPB2-5f2/RPB2-7cr [54,55], respectively. The sequences of ribosomal DNA internal transcribed spacer (*rDNA*
*ITS*), *EF-1α* and *RPB2* were amplified using primers ITS1/ITS4 [56], EF-728F/EF1LLErev [57] and fRPB2-5F/fRPB2-7cR [58] to identify *Trichoderma* species. PCR reaction was conducted in a total volume of 25 μL containing DNA template 1 μL, each primer 1 μL (10 μM), Taq PCR Mastermix (Sangon Biotech, Shanghai, China) 12.5 μL, and DNase free water 9.5 μL. The fragments of *rDNA ITS* were amplified with the conditions of 2 min at 94 °C, followed by 35 cycles of denaturation at 94 °C for 30 s, annealing at 58 °C for 30 s, initial extension at 72 °C for 30s, and remaining at 72 °C for 10 min. Amplification conditions for *EF-1α* and *RPB2* were 5 min at 94 °C, followed by 35 cycles of denaturation at 94 °C for 30 s, annealing at 55 °C for 30 s, initial extension at 72 °C for 1 min, and kept at 72 °C for 10 min. All primer sequences were listed in Appendix A. PCR products were checked by 1.5% agarose gel electrophoresis and sequenced using an ABI-PRISM3730 automatic sequencer (Applied Biosystems, Foster, CA, USA) in Sangon Biotech Co., Ltd. (Shanghai, China), 

### 4.5. Phylogenetic Analysis

Amplified sequences of *EF-1α* and *RPB2* genes from candidate *Fusarium* isolates were blasted on *Fusarium* MLST databases, while sequences of *rDNA ITS*, *EF-1α* and *RPB2* genes from candidate *Trichoderma* isolates were blasted on the NCBI database. Species of *Fusarium* and *Trichoderma* were identified based on the sequence similarity as compared to the reference species. For phylogenetic analysis, sequences of *Fusarium* and *Trichoderma* representative isolates, referred isolates and the outgroup isolate were edited and aligned with Clustalx 1.83, and characters were weighed equally. The referred isolates information of *Fusarium* and *Trichoderma* are listed in Appendix A. 

For *Fusarium* species, MEGA 5.0 (https://www.megasoftware.net/index.php, accessed on 14 May 2021) was used to calculate and analyze the differences in their base composition, and phylogenetic trees were constructed using neighbor-joining method based on both *EF-1α* and *RPB2* sequences for *Fusarium* spp., while those were performed for *Trichoderma* spp. based on either the combination of *rDNA ITS* and *EF-1α*, or the combination of *rDNA ITS* and *RPB2*, respectively. Clade support was inferred from 1000 bootstrap replicates, and alignment gaps were excluded. Novel sequence data were deposited in GenBank and the alignment in TreeBASE (www.treebase.org, accessed on 14 May 2021).

### 4.6. Pathogenicity Test of Fusarium Species

To test whether these *Fusarium* species identified from soybean rhizosphere are pathogenic, the representative isolates were randomly selected for pathogenicity test on the seedlings of soybean cultivar Nandou12, as described by Chang et al. (2020) [7].

### 4.7. In Vitro Antagonistic Effects of Trichoderma Strains on the Pathogenic F. oxysporum

For in vitro antagonistic assay, the representative strains of each *Trichoderma* species were randomly selected to test their antagonistic effects on the mycelium growth of *F. oxysporum*, a strong aggressive pathogen of soybean root rot. Both mycelial plugs of *F. oxysporum* and *Trichoderma* strains were inoculated on two opposite sides of 90 mm PDA plates with the distance of 75 mm, while only mycelial plug of *F. oxysporum* isolates inoculated on PDA plates were used as controls. Three plates were prepared for each isolate, and three independent experiments were conducted. All plates were incubated at 25 °C for 5 days in the darkness, and then the distance of *F. oxysporum* and *Trichoderma* on PDA plates was recorded. The percentage of mycelial inhibition (PMI) was calculated by the formula below (3).
(3)$$\mathrm{PMI}\%=\frac{\mathrm{the}\;\mathrm{colony}\;\mathrm{diameter}\;\mathrm{of}\;\mathrm{control}-\mathrm{the}\;\mathrm{colony}\;\mathrm{diameter}\;\mathrm{of}\;\mathrm{treat}}{\mathrm{the}\;\mathrm{colony}\;\mathrm{diameter}\;\mathrm{of}\;\mathrm{control}}\times 100$$

For 

For the visualization of hyphae interaction, on the confrontation culture, one sterile cover glass was inserted into PDA medium equidistant from *Trichoderma* and *F. oxysporum* at an angle of 45° and cultured at 28 °C in the dark. When both *Trichoderma* and *F. oxysporum* grew on the cover glass, the cover glass was taken out and observed under the compound microscope (Eclipse 80i, Nikon, Tokyo, Japan) and the mycelial interaction between *Trichoderma* and *F. oxysporum* was recorded and photographed.

### 4.8. Data Analysis

All data were recorded and processed using Microsoft office Excel 2010 (Microsoft Corporation, Redmond, Washington, USA). The data correlation was analyzed using generalized linear model (GLM) with quasi-poisson distribution for residuals, and statistical analysis was performed by Duncan’s test (*p* = 0.05) with GLM function using IBM SPSS Statistics 20 (IBM, Armonk, New York, USA). The composition proportion of *Fusarium* and *Trichoderma* spp. were calculated using the percentage of each species in total isolates from the corresponding soil sample, while isolation proportion of each species was evaluated using the percentage of each species in all isolates from four soil types. The difference in composition and isolation proportion of *Fusarium* or *Trichoderma* species in each cropping pattern was analyzed by Fisher’s exact test. 

## 5. Conclusions

In this study, maize–soybean relay strip intercropping increased the density of beneficial *Trichoderma* and decreased the pathogenic *Fusarium* communities (in rhizosphere diseased soil) as compared to soybean monoculture. Intercropping also affected species composition and isolation proportion of rhizosphere *Fusarium* and *Trichoderma* communities. The change in the pathogenic *Fusarium* might be primarily driven by the beneficial *Trichoderma* strains by mycoparasitism interaction. Overall, these findings suggest that maize–soybean relay strip intercropping alters both pathogenic *Fusarium* and beneficial *Trichoderma* communities in soybean rhizosphere soil and leads to antagonistic development of the fungal community structure in plant health, which regulates the suppressive effect of this intercropping on soybean root rot.

## Figures and Tables

**Figure 1 pathogens-11-00478-f001:**
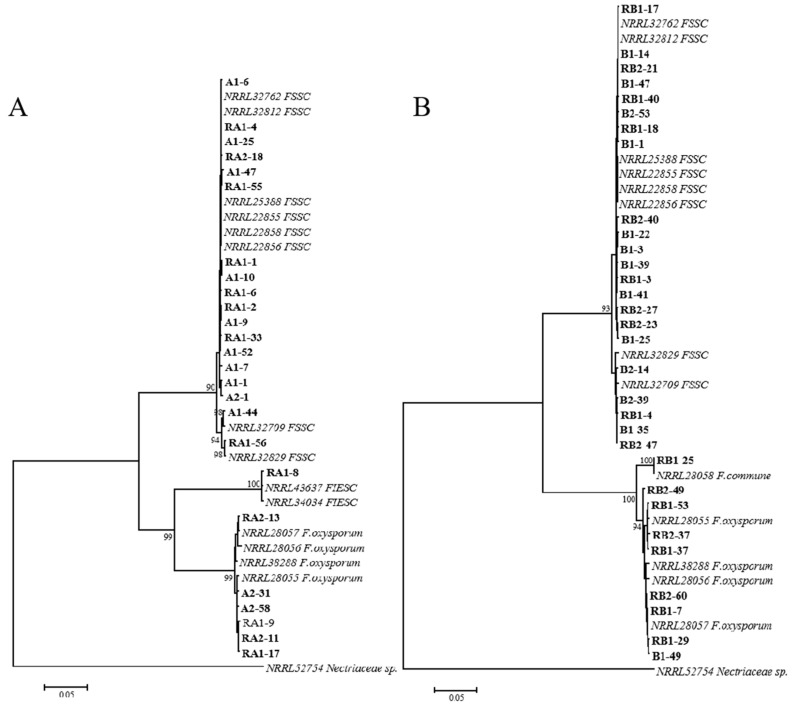
The phylogenetic tree based of *EF-1α* and *RPB2* genes was constructed using a neighbor-joining method for rhizosphere *Fusarium* isolates from intercropping and monoculture, respectively. (**A**) the phylogenetic tree constructed using *Fusarium* isolates from maize soybean relay strip intercropping. (**B**) the phylogenetic tree constructed using *Fusarium* isolates from soybean monoculture. Bootstrap support values were more than 90 % in both trees and obtained from 1000 replications. The *Nectriaceae* sp. (accession no. NRRL52754) was used as the outgroup isolate.

**Figure 2 pathogens-11-00478-f002:**
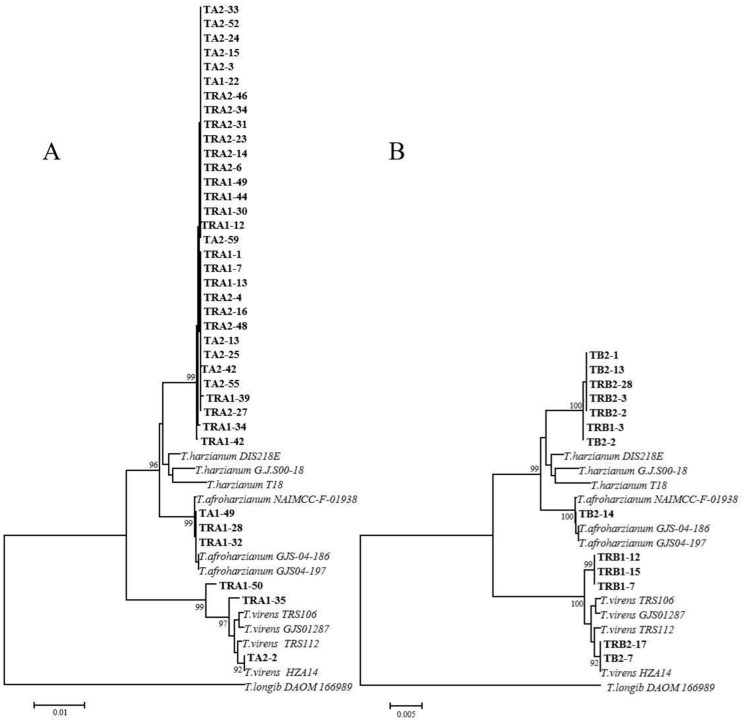
A neighbor-joining phylogenetic tree based on *ITS* and *RPB2* genes was constructed for intercropping and monoculture, respectively. (**A**) the phylogenetic tree constructed using *Fusarium* isolates from maize soybean relay strip intercropping. (**B**) the phylogenetic tree constructed using *Fusarium* isolates from soybean monoculture. Bootstrap support values were more than 92% in both trees and obtained from 1000 replications. The *Trichoderma longib* (DAOM 166989) was used as the outgroup isolate.

**Figure 3 pathogens-11-00478-f003:**
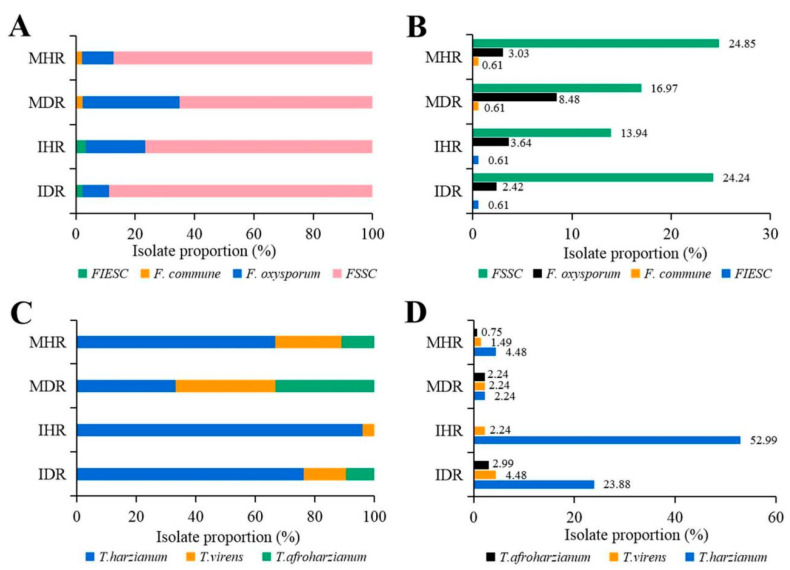
(**A**) Composition proportion of each *Fusarium* species in each soil sample. (**B**) Isolation proportion of each *Fusarium* species in total *Fusarium* isolates from all soil samples. (**C**) Composition proportion of each *Trichoderma* species in each soil sample. (**D**) Isolation proportion of each *Trichoderma* species in total *Trichoderma* isolates from all soil samples. The difference in composition percentage and isolation percentage was analyzed by Fisher’s exact test.

**Figure 4 pathogens-11-00478-f004:**
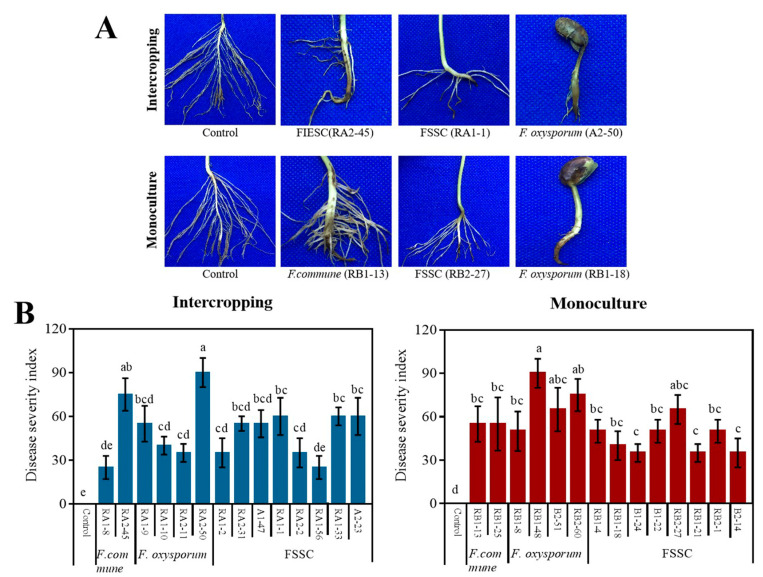
(**A**) Symptoms of soybean seedlings after inoculation with representative isolates of rhizosphere *Fusarium* species from intercropping and monoculture; FIESC (RA2-45), FSSC (RA-1), *F. oxysporum* (A2-50), *F. commune* (RB1-13), FSSC (RB2-27) and *F. oxysporum* (RB1-18) were the representative isolates of the corresponding *Fusarium* species from intercropping and monoculture. Control means soybean seedings without *Fusarium* inoculation. (**B**) Disease severity index of soybean root rot caused by representative *Fusarium* isolates in both cropping patterns. Difference significance was calculated from three independent replicates using SPSS 20 software at the level of *p* = 0.05 and marked by lowercase.

**Figure 5 pathogens-11-00478-f005:**
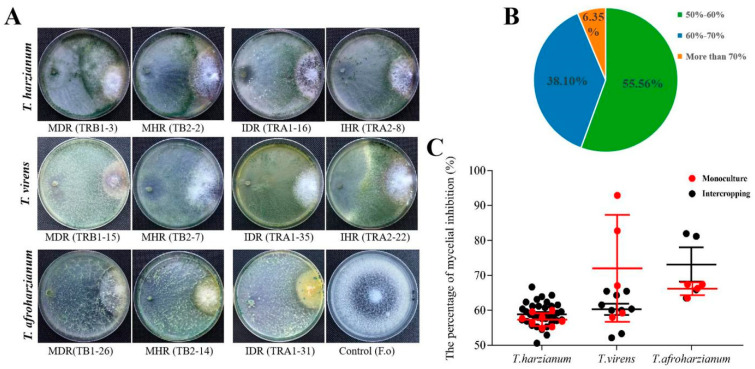
(**A**) Confrontation culture of *Trichoderma* isolates against *F. oxysporum* on PDA plates. IDR and IHR are the abbreviations of diseased rhizosphere soil and healthy rhizosphere soil in maize–soybean relay strip intercropping, respectively; MDR and MHR stand for rhizosphere diseased soil and rhizosphere healthy soil in soybean monoculture, respectively. The isolate codes are marked in the brackets following the abbreviation of soil samples. (**B**) Statistical analysis of the percentages of *Trichoderma* isolates with different inhibition of mycelial percentage (IP) of 50–60%, 60–70% and more than 70%, respectively. (**C**) The inhibition percentage of *Trichoderma* isolates for three species against *F. oxysporum* in both maize–soybean relay strip intercropping and monoculture which are marked as black dots and red dots in the graph, respectively. The error bar represents the standard error of the mean value of three independent replicates, and the graph is drawn by GraPhpad Prism 7.00 software (https://www.graphpad.com/ accessed on 14 May 2021).

**Figure 6 pathogens-11-00478-f006:**
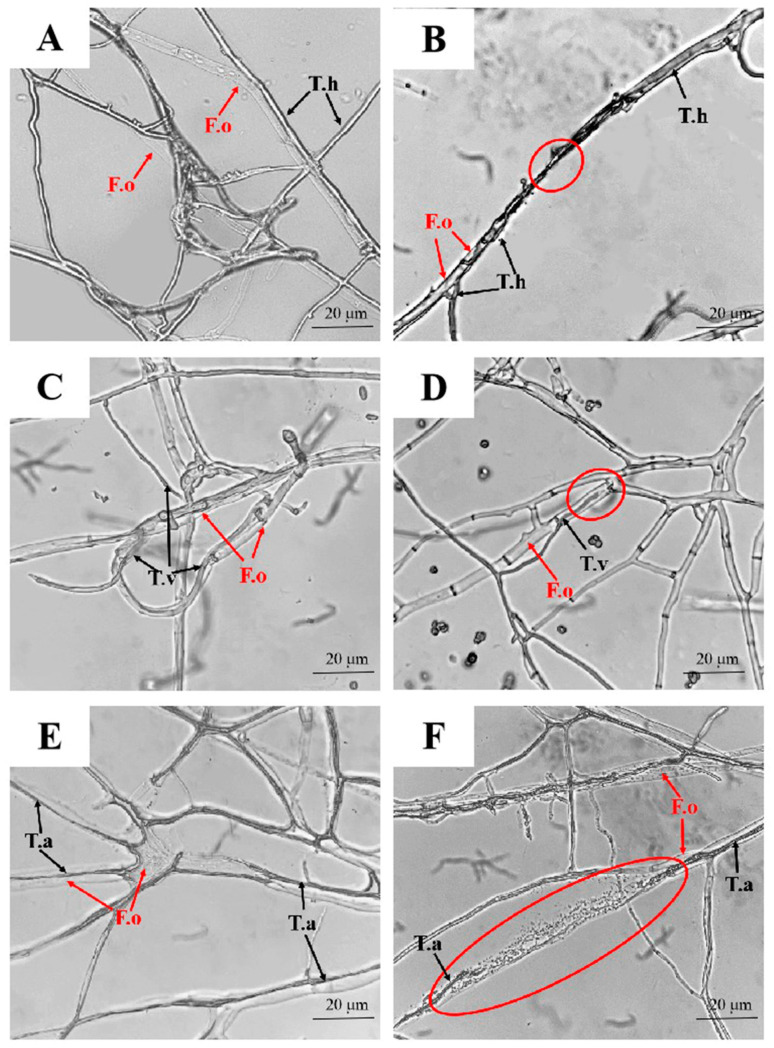
The hyphae interaction of *T. harzianum* (**A**,**B**), *T. virens* (**C**,**D**) and *T. afroharzianum* (**E**,**F**) with *F. oxysporum* was observed under a compound microscope after confrontation culture on PDA plates for 3 days. The area marked by red circle means the interaction points of *Trichoderma* spp. and *F. oxysporum*. The hyphae of *T. harzianum*, *T. afroharzianum*, *T. virens* and *F. oxysporum* are abbreviated as T.h, T.a, T.v and F.o, respectively, and marked by red or black arrows. The scale bar represents 20 µm.

**Table 1 pathogens-11-00478-t001:** Population richness of *Fusarium* and *Trichoderma* communities in different soil samples from intercropping and soybean monoculture.

Cropping Patterns	Soil Samples	Sample Code	*Trichoderma* Richness (cfu·g^−1^)	*Fusarium* Richness (cfu·g^−1^)
Intercropping	Diseased rhizosphere soil	IDR	2560.98 ± 2414.51 ab	1046.51 ± 164.44 a
	Healthy rhizosphere soil	IHR	4404.76 ± 168.36 a	329.67 ± 186.49 b
Monoculture	Diseased rhizosphere soil	MDR	523.26 ± 82.22 b	391.62 ± 12.88 b
	Healthy rhizosphere soil	MHR	523.26 ± 82.22 b	458.09 ± 151.62 b

Notes: Different lowercase indicate the significant difference at the level of *p* = 0.05 using SPSS ANOVN.

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
