# Peer review of "Changes in the Density and Composition of Rhizosphere Pathogenic Fusarium and Beneficial Trichoderma Contributing to Reduced Root Rot of Intercropped Soybean"

_pathogens, 2022, doi:10.3390/pathogens11040478_

Round 1

Reviewer 1 Report

The study addresses a topic of great interest, is based on a large data set, and reports a number of results that are significant for further basic as well as applied research.  However, the text is written in a somewhat confusing style, including a number of run-on sentences, and needs thorough review by a native English speaker. Here are some examples:

line 20- Instead of "Maize-soybean intercropping can suppress soybean root rot....." Maize-soybean intercropping has been shown to significantly control the type of soybean root rot that tends to occur in monoculture."

line 42- Instead of "It is often infected by a variety ....." A variety of pathogens, including Fusarium, cause soybean root rot, leading to 10-60%  loss of soybean yield."

line 74- Instead of "Accumulating researches indicate....." Research increasingly indicates....."

Author Response

The study addresses a topic of great interest, is based on a large data set, and reports a number of results that are significant for further basic as well as applied research.  However, the text is written in a somewhat confusing style, including a number of run-on sentences, and needs thorough review by a native English speaker. Here are some examples:

Response: Thank you very much for your positive evaluation on our manuscript. We have invited one expert of English language to help us polish our manuscript expression. We hope our revised manuscript is suitable for publication in Pathogens.

line 20- Instead of "Maize-soybean intercropping can suppress soybean root rot....." Maize-soybean intercropping has been shown to significantly control the type of soybean root rot that tends to occur in monoculture."

Response: According to reviewer’s suggestion, we have modified the sentence as “Maize-soybean intercropping has been shown to significantly control the type of soybean root rot that tends to occur in monoculture.” instead of “Maize-soybean intercropping can suppress soybean root rot.....”

line 42- Instead of "It is often infected by a variety ....." "

Response: Thank you a lot. We have used “A variety of pathogens, including Fusarium, cause soybean root rot, leading to 10-60% loss of soybean yield.” instead of “It is often infected by a variety .....”

line 74- Instead of "Accumulating researches indicate....." Research increasingly indicates....."

Response: We have changed “Accumulating researches indicate.....” by " Research increasingly indicates....."

Reviewer 2 Report

The manuscript is well structured and the results presented are interesting. Some misspellings in the attached file were highlighted. I suggest separating sentences that are too long, particularly in the results and discussion.

Author Response

The manuscript is well structured and the results presented are interesting. Some misspellings in the attached file were highlighted. I suggest separating sentences that are too long, particularly in the results and discussion.

Response: Thank you very much for your positive evaluation on our manuscript. We have invited one expert of English language to help us polish our manuscript expression. We hope our revised manuscript is suitable for publication in Pathogens.

In addition, according to reviewers kind and careful suggestions, we have also rewritten our long sentences and revised some other errors or improper expression in the highlight part of our revised manuscript.

Reviewer 3 Report

This manuscript by Xu et al. provides Maize-soybean relay stripe intercropping increased Trichoderma density while lowering Fusarium sp. density when compared to soybean monoculture. The experiments were well organized and conducted precisely, and the data logically support the conclusion as stated in the text. Based on the above observations, they performed commonly used standard assays to demonstrate the intercropping suppress the pathogenic Fusarium communities. However, some major issues that need to be addressed before acceptance.

Comments

  1. There are several typographical problems in this MS, which should be corrected (Ex. Line No. 247, 426, 428, etc).
  2. Line 414: should provide the reference's first author's name?
  3. Which year, season, and time period were the samples collected?
  4. To prove advantageous Trichoderma, the expression of defense-related genes (root rot) in soybean plants should be investigated.
  5. Authors should improve the quality of the figures.
  6. The discussion should be expanded to include the topic of Fusarium suppression.

Author Response

This manuscript by Xu et al. provides Maize-soybean relay stripe intercropping increased Trichoderma density while lowering Fusarium sp. density when compared to soybean monoculture. The experiments were well organized and conducted precisely, and the data logically support the conclusion as stated in the text. Based on the above observations, they performed commonly used standard assays to demonstrate the intercropping suppress the pathogenic Fusarium communities. However, some major issues that need to be addressed before acceptance.

Comments

  1. There are several typographical problems in this MS, which should be corrected (Ex. Line No. 247, 426, 428, etc).

Response: Thank you for your reminding. We have check these typographical problems and corrected them. We also checked the whole manuscript and tried our best to remove these kinds of problems.

  1. Line 414: should provide the reference's first author's name?

Response: We are sorry for this mistake. In our revised manuscript, we have added the reference’s first authors name in this sentence “...the seedling of soybean cultivar Nandou12 as described by Chang et al. (2020) [7].”

  1. Which year, season, and time period were the samples collected?

Response: The soil samples were collected at R2 growth stages of soybean in summer of 2018, and we also added these information in parts of our material and methods.

  1. To prove advantageous Trichoderma, the expression of defense-related genes (root rot) in soybean plants should be investigated.

Response: Thank you for your advice. Actually, in our another manuscript we tested the expression of PR1, PR5, LOX genes involving in SA or JA signaling pathways, and several genes involving in isoflavonoid biosynthesis pathway aiming to uncover the antagonism mechanism of four Trichoderma with high inhibition effect on Fusarium causing soybean root rot. And we also found that the control effect of Fusarium root rot with the combined treatments of four Trichoderma was much better than that treatment of single isolate Trichoderma.

  1. Authors should improve the quality of the figures.

Response: Thank you. We have tried our best to modify our manuscript and replaced our figures with high resolution, such as Figure 5.

  1. The discussion should be expanded to include the topic of Fusarium

Response: Thanks a lot. In our study, we found that Fusarium community was significantly suppressed in the healthy rhizopshere soil as compare to diseased rhizosphere soil of intercropping, and this was followed by a remarkable increase of Trichoderma community, indicating that Trichoderma might interact with Fusarium. To prove this hypothesis, in-vitro hyphae inhibition of F. oxysporum was tested after treatment with Trichoderma strains, and hyphae interaction of both fungi was also observed under the microscope. We added the discussion on the Fusarium suppression in our revised manuscript.

Round 2

Reviewer 3 Report

As requested, the authors significantly improved the manuscript.